# A Green and Cost-Effective Chromatographic Method for the Measurement of the Activity of Human Serum Carnosinase

**Chiara de Giacomi and Luca Regazzoni ***

Department of Pharmaceutical Sciences, University of Milan, Via Mangiagalli 25, 20133 Milan, Italy
* Correspondence: luca.regazzoni@unimi.it

**Abstract:** Carnosinase (i.e., CN1; E.C. 3.4.13.20) is an enzyme found in the sera of higher primates. CN1 preferentially catalyzes the hydrolysis of natural, orally adsorbed histidine dipeptides like carnosine (i.e., β-alanyl-L-histidine). This is the reason why carnosine has a limited use as a human food supplement or pharmacological agent, despite the promising results obtained in experiments on animal models of human diseases. Herein, an assay is reported for the measurement of serum CN1 activity. The method is intended for the screening of CN1 inhibitors able to enhance carnosine bioavailability in humans. The method was developed to monitor serum hydrolytic activity via the quantitation of one of the products of carnosine hydrolysis (i.e., histidine). Separation was achieved without using organic solvents by means of ion chromatography (IC), whereas detection was provided by UV spectroscopy. The assay herein reported is a green and cost-effective alternative to a recently published method based on hydrophilic interaction liquid chromatography (HILIC) and mass spectrometry (MS). The results show that such a method produces reliable measurements of serum hydrolytic activity and can be used for the screening of CN1 inhibitors.

**Keywords:** carnosine; human serum carnosinase; bestatin; enzyme inhibition; ion chromatography; UV detection





## 1. Introduction

Human carnosinase (i.e., CN1; E.C. 3.4.13.20) is a zinc-dependent, homodimeric enzyme belonging to the M20 family of metalloproteases [1,2]. In humans and higher primates, CN1 is produced by the brain, but it is found also in serum since it reaches the bloodstream through the cerebrospinal fluid [2,3].

CN1 preferentially catalyzes the hydrolysis of dipeptides like carnosine (i.e., β-alanyl-L-histidine) or other dipeptides with a histidine at C-terminus. However, dipeptides with a methylated imidazole ring (e.g., anserine and balenine) or with other amino acids at the C-terminus can also be hydrolyzed by CN1, although the hydrolysis of such substrates is slower than the hydrolysis rate of carnosine [3,4].

Serum hydrolytic activity towards carnosine has quite a broad interindividual variability depending on CN1 gene polymorphism [5], CN1 serum content [6] and age [7]. Modifications of CN1 activity are associated with several human diseases [8]. Interindividual differences correlate with onset risk for some human disorders like diabetic nephropathy. Specifically, low CN1 activity is associated with a reduced risk of developing diabetic nephropathy [5]. A potential explanation for such an effect is that reduced CN1 activity allows the accumulation of carnosine in the kidney, which can exert a protective role against renal damages due to glucose accumulation. In fact, carnosine has buffering and metal-chelating properties and is able to protect against the damage induced by oxidants or electrophiles such as reactive carbonyl species [9].

Although there are human disorders that are characterized by CN1 deficiency and associated with high concentrations of peripheral carnosine [10,11], the adverse effects on health are probably the consequence of the CN1 deficiency in the brain and not of the

accumulation of carnosine in the blood (i.e., carnosinemia). This hypothesis is sustained by human studies in which carnosinemia was transiently promoted in humans, without observing the adverse effects found in CN1 deficient subjects [12]. Such studies were performed since a number of animal studies showed the benefits of carnosine supplementation for the mitigation of atherosclerosis [13,14], diabetes [15], metabolic syndrome [16,17] and neurological disorders [18]. However, the benefits of carnosine supplementation observed in rodents cannot be fully replicated in humans since serum CN1 activity limits carnosine bioavailability and its distribution to peripheral tissue via blood circulation.

Since carnosine has potential applications as a food supplement or a drug, some studies have been focused on the development of bioavailable derivatives designed to be CN1-resistant (e.g., carnosinol and D-carnosine) [17,19]. Some of these derivatives were still poorly available since the structural modification of carnosine had a negative impact on other features that are fundamental for bioavailability (i.e., oral absorption). Therefore, prodrugs were also designed to improve the bioavailability of derivatives that were CN1-stable but poorly absorbed [14,20]. Alternatively, some authors reported the use of CN1 inhibitors as an alternative strategy to improve carnosine bioavailability. Carnostatine was reported as a hit compound from the screening of a library of molecules [21]. Such a screening was performed by measuring the hydrolysis rate of carnosine to identify compounds able to interfere with CN1 activity. The analytical method used to this scope relies on the quantitation of the amount of histidine produced over time from carnosine hydrolysis. To allow the detection of histidine by fluorescence, a derivatization step with o-phtaladehyde (OPA) was, however, necessary [7]. The limits of OPA derivatization have been recently highlighted by developing an alternative approach based on hydrophilic interaction liquid chromatography with mass spectrometric detection (HILIC-MS). The method allowed direct detection and evidenced that derivatization gave reliable results only for some CN1 substrates [4]. However, although the HILIC-MS method result is more reliable than that of OPA-based approaches, it is less cost-effective.

For this reason, the setup and validation of an alternative method based on ion chromatography with ultraviolet spectrophotometric detection (IC-UV) is herein reported. Such a method is green and cost-effective compared to HILIC-MS, and the data herein reported demonstrate that it provides an accurate measurement of serum hydrolytic activity towards carnosine while retaining all the advantages of the HILIC-MS method. Specifically, the IC-UV method allows the direct detection of the analytes, without derivatization. Like for HILIC-MS, the IC-UV method also has a flexible setup and can be easily adapted to experiments based on the use of different CN1 substrates (i.e., carnosine and homocarnosine). Finally, the IC-UV method was also successfully used for studies on CN1 inhibition, and therefore, it is potentially applicable to the screening and testing of molecules designed to inhibit CN1.

## 2. Materials and Methods

### 2.1. Chemicals

HPLC-grade water (18 MΩ × cm, 25 °C) was purified with a Milli-Q water system (Millipore; Milan; Italy). HPLC-grade solvents were purchased from Scharlab (Lodi, Italy). Pooled human serum from AB donors, potassium chloride, hydrochloric acid, histidine hydrochloride, L-homocarnosine sulfate, bestatin hydrochloride and all the other chemicals were purchased from Sigma-Aldrich (Merck Life Science, Milan, Italy). Carnosine was provided by Flamma s.p.a. (Chignolo D'isola, Bergamo, Italy).

### 2.2. Sample Preparation

#### 2.2.1. Stock Solutions and Standards

Carnosine, homocarnosine, histidine chloride, bestatin chloride and every other compound used for serum experiments (see Section 2.2.3) were dissolved in water to obtain stock solutions at a 5 mM concentration. Before incubation with serum, working solutions

were prepared by further dilution of the stock solutions with freshly prepared 10 mM phosphate buffer, pH 7.4.

Calibration curve standards were prepared in triplicates by diluting histidine stock solutions in water down to concentrations ranging from 1 to 10 µM. All calibration curve standards were fortified with the same amount of carnosine (i.e., 5 µM final concentration).

Three sets of histidine reference standards (HRS) were prepared to estimate recovery and matrix effect. HRSs were prepared by using three different matrices fortified with the same amount of histidine. A first set (HRS1, 6 samples) was prepared in water; a second set (HRS2, 6 samples) was prepared in supernatants obtained from deproteinization of human serum (see Section 2.2.2); and a third set (HRS3, 6 samples) was prepared in serum. HRSs prepared in serum were deproteinized right after being fortified with carnosine to minimize analyte loss due to CN1 hydrolysis.

The working solutions and standards were filtered through Millex®-HV membranes before the analyses (13 mm filter diameter, 0.45 µm membrane, Merck Millipore S.p.A. Vimodrone, Italia).

### 2.2.2. Serum Protein Precipitation

Aliquots of serum were diluted 10-fold with ethanol and left at 4 °C for 10 min. Samples were then centrifuged at 14,000 RPM, and supernatants were collected in glass autosampler vials and further diluted by addition of an equal volume of water to reduce the ethanol concentration to below 50%.

### 2.2.3. Determination of Serum Hydrolytic Activity

The procedure resembled the protocol reported by Gilardoni et. al, with minor modifications [4]. Serum aliquots were diluted fivefold with 10 mM phosphate buffer, pH 7.4. For inhibition experiments, the serum was diluted fivefold with working solutions of bestatin, prepared at different concentrations ($10^{-3}$–$10^3$ µM range). Hydrolysis kinetics were started by mixing equal volumes of diluted serum and a working solution of the substrate (i.e., 100 µM carnosine or homocarnosine). Corresponding control samples were prepared by mixing equal volumes of diluted serum and 10 mM phosphate buffer, pH 7.4. The aliquots of diluted serum, phosphate buffer and working solutions used for kinetics experiments were prewarmed at 37 °C for 5 min before being mixed. Kinetics were performed at 37 °C and stopped by precipitating serum proteins (see Section 2.2.2) at the desired time (i.e., 15 min for carnosine and 3 h for homocarnosine). Aliquots containing 20 µL of the supernatants obtained from protein precipitation were sampled and analyzed as described in Section 2.3.

### 2.3. Chromatography (IC-UV Method)

The analyses were provided by a Surveyor HPLC system (Thermo Fisher Scientific, Rodano, Milan, Italy) equipped with a PolySULFOETHYL A™ ion chromatography (IC) column (200 × 4.5 mm, 5 µM particle size, 300 Å pore size, PolyLC INC., Columbia, MD, USA).

The elution of the analytes was provided by an 800 µL/min flow rate consisting of 15 mM aqueous KCl adjusted with hydrochloric acid down to pH = 3. Samples were stored into the autosampler tray at 10 °C throughout the analyses, whereas the column compartment was kept at 30 °C. An Optisolv mini filter of 0.5 µm (Sigma Aldrich, Merck Life Science, Milan, Italy) was installed to the column head to protect it from particulates.

Ultraviolet (UV) spectrophotometric detection within a wavelength range from 200 to 400 nm was provided by a photodiode array detector. Instrument control was provided by the software Xcalibur 2.0.7 (Thermo Fisher Scientific, Rodano, MI, Italy).

### 2.4. Data Analysis and Statistics

Peak areas were extracted from single-wavelength chromatograms (220 nm). Peak area extraction was performed by using the Genesis algorithm with valley detection feature

activated and default parameters for pick picking (Qual browser utility of Xcalibur 2.0.7, Thermo Fisher Scientific, Rodano, MI, Italy).

Peak resolution (Rs) was calculated as recommended from IUPAC [22]. The experimental Rs was compared to the guideline target value for system suitability (i.e., Rs = 2). Comparison was performed by using one sample t-test upon the verification of data normality using the Shapiro–Wilk normality test. Data that were not normally distributed were compared by the nonparametric Wilcoxon signed-rank test.

The calibration curve for histidine quantitation was fitted by using the simplest model describing the relationship between the amount of histidine injected and the area of its chromatographic peak.

Intermediate precision and accuracy were evaluated by analyzing four independent samples. Intermediate precision was measured as the relative standard deviation of the measures, whereas accuracy was measured as the relative standard error between the nominal concentration of the samples and the concentration calculated using the calibration curve.

Ordinary one-way ANOVA with a Sidak multiple comparison test was applied for the comparison of the area of histidine peaks obtained by analyzing standards prepared in different matrices (HRS, see Section 2.2.1). An equivalent nonparametric test (i.e., Kruskal–Wallis test with Dunn's multiple comparison test) was used in case data did not pass the Shapiro–Wilk normality test. Matrix effect (ME) and histidine recovery (HR) were then measured as follows:

$$ME = area\ HRS2/area\ HRS1 \times 100 \tag{1}$$

$$HR = area\ HRS3/area\ HRS2 \times 100 \tag{2}$$

Since ME and HR were ratios of two independent series of measures, the average values and standard deviations were calculated by approximations of second-order Taylor series expansions, as reported in the literature [23].

The serum hydrolysis rate (SHR) was calculated by considering the nanomoles of histidine produced in the reaction batch (nmol $_{His}$), the volume of non-diluted serum added in the reaction batch (V) and incubation time before deproteinization (i.e., kinetics endpoint, t), μ:

$$SHR = nmol\ _{His}/(V*t), \tag{3}$$

Inhibition curves were fitted by using the simplest model describing the relationship between the residual SHR and the concentration of bestatin added in human serum, starting from a dose–response inhibition curve without variable slope.

Fitting models (i.e., histidine calibration curve and bestatin inhibition curve) were compared using the extra-sum-of-squares F test to select the most appropriate model that fits the experimental data. Curve fitting and all the statistical tests mentioned were performed using Prism (version 9, GraphPad software LLC, Boston, MA, USA).

## 3. Results and Discussion

As in Figure 1, when an aqueous solution containing equimolar amounts of histidine and carnosine is analyzed using the IC-UV method reported in Section 2.3, two peaks are detectable at 220 nm.

The analysis of samples containing pure histidine or carnosine allowed us to identify the peaks at 10.5 and 11.5 min as histidine, and carnosine, respectively.

The separation was quite reproducible since the variability of the retention time of the analytes was 2.4% for histidine and 2.1% for carnosine, with an average Rs of 1.2 (relative standard deviations measured on six independent samples, with 3.2% experimental variability). The Wilcoxon signed-rank test rejected with 96.88 percent of confidence the null hypothesis that such a value tops the recommended value (i.e., Rs = 2) for passing a system suitability test required by some method validation guidelines [24]. Furthermore, the resolution was not improvable by modifications of the flow rate (0.7–1.5 mL/min

range), of the column temperature (25–50 °C range), of the mobile phase additives (i.e., NaCl instead of KCl) or of the eluent pH (2–7 range).

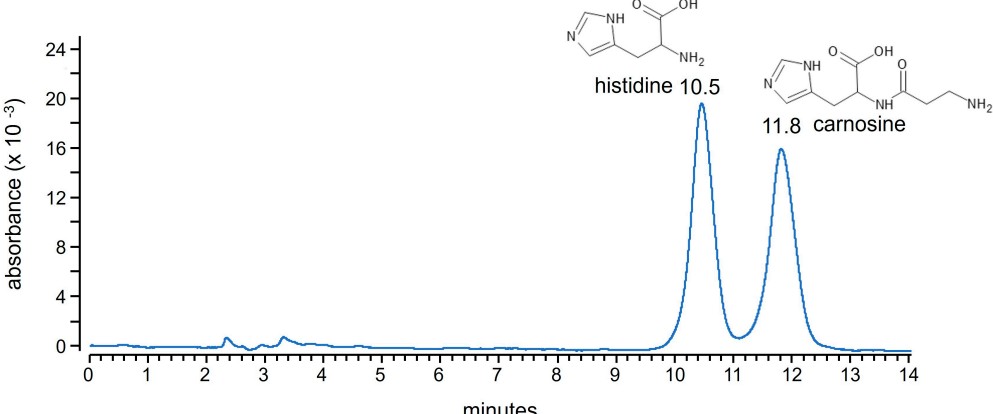

**Figure 1.** IC-UV analysis of an aqueous solution of histidine and carnosine (both analytes are 10 µM).

Although the separation was not compliant with the recommended Rs value, the method was looking suitably convenient for the setup of a green and cost-effective chromatographic assay for the measurement of serum hydrolytic activity towards carnosine. Specifically, a chromatographic method intended for this scope was recently reported by Gilardoni et al. [4]. The method was operated with hydrophilic interaction chromatography coupled with mass spectrometric detection (HILIC-MS). However, an IC-UV separation has several apparent advantages over HILIC-MS. First, HILIC chromatography requires the use of acetonitrile as the main eluent. On the contrary, IC separation is provided by water containing a low millimolar concentration of inorganic salts. Second, the method reported in Section 2.3 relies on the use of a UV detector instead of MS. These two features make the IC-UV method green and cost-effective compared to the HILIC-MS method. Finally, the HILIC-MS method measures the serum hydrolysis rate (SHR) based on the quantification of the residual amount of the substrate (e.g., carnosine, anserine, balenine or homocarnosine) [4]. Conversely, the IC-UV method potentially allows the measurement of the SHR via the quantification of one of the products of carnosine hydrolysis (i.e., histidine) with no interference from the substrate. The first requirement of such an assay is, however, that IC-UV must provide sufficient sensitivity to quantitate the amount of histidine produced. According to the procedure reported in Section 2.3, a sample preparation and an SHR similar to those reported for the HILIC-MS method [4] are expected to undergo an injection of 50 picomoles of histidine per run.

Therefore, histidine standards were prepared and analyzed to provide the injection of histidine in amounts ranging from 20 to 140 picomoles. All the histidine standards were fortified to have a final amount of 100 picomoles of carnosine injected along with histidine.

This procedure was necessary since the measurement of the SHR requires us to stop the kinetics before the complete hydrolysis of the substrate [4]. Histidine quantitation must be therefore provided with a residual trace of carnosine in the sample.

Despite an Rs of 1.2, the relationship between the histidine concentration and the area of its chromatographic peak was linear (adjusted R square: 0.98; see Figure 2).

Both precision and accuracy remain within 15% for all the standards (see Table 1).

The IC-UV method can therefore give a reliable quantitation of histidine down to 20 picomoles injected, despite the presence of carnosine and a resolution not compliant with the recommended Rs target for method validation.

Although such data suggest that the method has sufficient sensitivity, the sample preparation procedure necessary for the measurement of the SHR by means of HILIC-MS could not be applied to IC-UV without adjustments. Specifically, a tenfold dilution in acetonitrile could not be used to induce protein precipitation and stop carnosine hydrolysis because of its negative impact on the Rs. Standards of histidine and carnosine prepared

in aqueous acetonitrile and analyzed by IC-UV resulted in broad, unretained and poorly resolved peaks. Trichloroacetic or perchloric acid at the typical concentrations used for deproteinization were also detrimental to the Rs. On the contrary, a tenfold dilution of serum with ethanol was as effective as acetonitrile for protein precipitation, with no impact on separation after a dilution with water to reduce ethanol content to below 50% (see Figure 3).

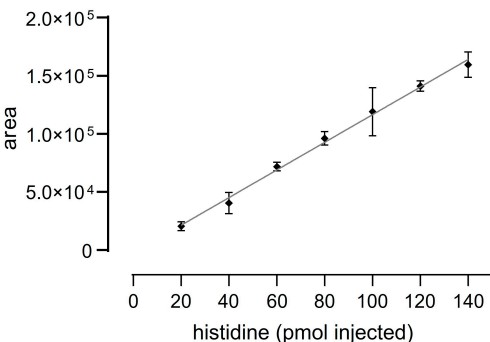

**Figure 2.** Calibration curve obtained by the analysis of histidine standards prepared in water (areas reported as means ± standard deviations).

**Table 1.** Precision and accuracy (N = 4) of IC-UV method.

| Histidine (Picomoles Injected) | Precision (Relative Standard Deviation) | Accuracy (Relative Standard Error) |
|---|---|---|
| 20 | 12.1% | −3.9% |
| 40 | 14.1% | −9.6% |
| 60 | 3.3% | 4.1% |
| 80 | 3.9% | 3.7% |
| 100 | 11.0% | 2.3% |
| 120 | 2.0% | 0.6% |
| 140 | 4.4% | −2.7% |

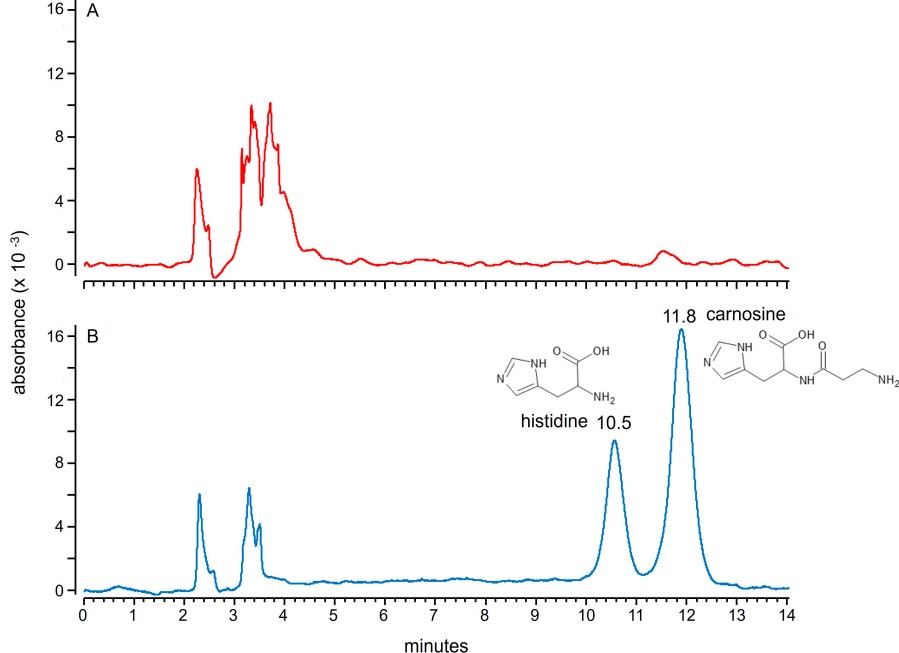

**Figure 3.** IC-UV analysis of two pooled human serum samples upon protein precipitation. Sample (**A**) (red line) is pooled human serum; sample (**B**) (blue line) is pooled human serum fortified with histidine and carnosine (25 μM and 50 μM, respectively).

The main UV-detectable serum interferents that were not removed by deproteinization eluted within 5 min and did not overlap with carnosine or histidine peaks. Although several studies in literature report that human serum is expected to contain histidine in a wide range of concentration [25–28], the commercially available pooled serum used for the experiments had undetectable amounts of histidine (see Figure 3). The most likely explanation for the absence of histidine is that the overall serum dilution necessary for sample preparation (200-fold dilution) decreases the histidine concentration below the limit of detection of the IC-UV method. Notably, carnosine was also not detectable in serum (see Figure 3), although this is expected owing to the activity of CN1.

These findings are analytically appealing since the SHR can be measured by analyzing only one sample at the incubation endpoint to determine the amount of histidine produced by carnosine hydrolysis. On the contrary, the assay based on HILIC-MS requires the quantification of the residual amount of carnosine via the analysis of two samples: one collected at the beginning of the incubation and a second one collected at the kinetics endpoint [4]. The amount of hydrolyzed carnosine can be then calculated from HILIC-MS data as the difference of the amount of carnosine detected at the beginning and at the endpoint of the incubation. This procedure has the risk of the propagation of analytical errors and is less appealing than a direct measurement of the amount of histidine produced by carnosine hydrolysis.

When commercially available pooled serum was spiked with carnosine and then analyzed by IC-UV, the residual amount of carnosine decreased over time, with a simultaneous increase in the peak of histidine (see Figure 4).

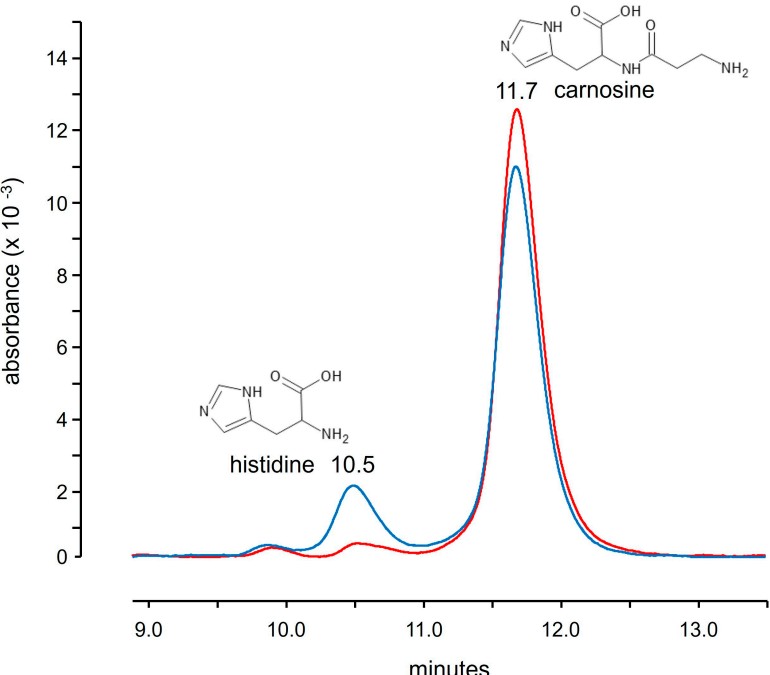

**Figure 4.** IC-UV analysis of pooled human serum samples fortified with 50 μM of carnosine upon protein precipitation. The red line is the chromatogram obtained right after serum fortification; the blue line is the chromatogram obtained after 10 min of serum fortification.

Trace levels of histidine were detectable in samples analyzed right after spiking carnosine into the serum. However, the peak was not detectable without spiking carnosine, so such an amount of histidine can be generated by the rapid hydrolysis of carnosine during the time required for picking up an aliquot of serum from the incubation batch and transferring it to another container for protein precipitation and analysis. The quantification of the amount of histidine produced by carnosine hydrolysis was performed by using the calibration curve obtained in water (see Figure 3). In fact, the matrix effect was negligible,

and analyte recovery was complete. This was demonstrated by ordinary one-way ANOVA with a Sidak multiple comparison test applied for the comparison of the peak areas of 18 independent standards fortified with histidine (HRS, see Section 2.2.1). The standards were prepared in three different matrices (i.e., water, serum or supernatant obtained from serum deproteinization), but the statistical test rejected the null hypothesis that histidine peak areas were different based on the matrix used. Concerning the matrix effect, the average area of histidine peaks in serum supernatants was $105.2 \pm 5.9\%$ of the average values obtained in water. For histidine recovery, the average area of histidine peaks in serum samples was $103.1 \pm 4.4\%$ of the average values obtained in serum supernatants.

Twelve samples of pooled human serum fortified with carnosine were then analyzed on different days across one month to determine the SHR. An average value of 1.14 nmol per hour per microliter of serum with a relative standard deviation of 19% was obtained from the analyses (range of the measures: 0.9–1.6).

These numbers are consistent with the activity range reported in the literature (i.e., 0.9–1.3). Such a range was found by using serum samples from single donors, which were fortified by using a single concentration of carnosine (80 μM) [29]. However, the measurement of the SHR was affected by the experimental conditions since the range was extended (i.e., from 0.9 to 3.1) when different concentrations of carnosine were used for the measurements [29]. The results obtained by IC-UV were compared to the data reported by using HILIC-MS since the IC-UV method was developed to resemble such a method (i.e., both methods used the same concentration of carnosine to fortify the serum, the same dilution of human serum, a similar incubation time and similar deproteinization protocols, and both methods was performed on commercially available pooled human serum).

An SHR of $1.3 \pm 0.1$ nmol per hour per microliter of serum was measured by HILIC-MS on commercially available pooled human serum [4], but a two-tailed, unpaired t test rejected with 95% confidence the null hypothesis that the IC-UV method produces different results. As for HILIC-MS, IC-UV was successfully applied for the measurement of the hydrolytic activity towards homocarnosine. Homocarnosine is another CN1 substrate that produces histidine upon its hydrolysis. Since the IC-UV assay is based on histidine quantitation, the measurement of the homocarnosine hydrolysis rate was performed just by changing the substrate spiked into serum (i.e., homocarnosine instead of carnosine) and the incubation time (i.e., 3 h of incubation instead of 15 min). An extension of the incubation time was necessary to produce quantifiable amounts of histidine. In fact, the homocarnosine hydrolysis rate is about two orders of magnitude slower than the carnosine hydrolysis rate [3,4]. An average value of 0.015 nmol per hour per microliter of serum (range: 0.011–0.019) with a relative standard deviation of 26% was obtained when homocarnosine was used as a substrate. As for the measurement performed with carnosine, such a result was compared with the value reported in the literature using the HILIC-MS method. The unpaired t test rejected with 95% confidence the null hypothesis that the two methods produce different measurements. Therefore, IC-UV can be considered a reliable assay since it was validated against a published alternative method on different substrates of CN1.

Although the Rs was a concern when we first tested the method for the measurement of the hydrolysis of carnosine (or homocarnosine) operated by CN1, the results were consistent with the literature. This means that the method has a sufficient Rs to accomplish the intended purpose, regardless of the method used to measure the Rs or its minimum acceptable value, according to various guidelines.

To explore further applications of the IC-UV method, experiments on serum incubated with bestatin were performed to assess whether the assay is applicable to the screening of CN1 inhibitors. Bestatin is a known inhibitor of CN1 [1], and preliminary analyses established that such a compound elutes within 5 min and does not interfere with histidine quantitation by IC-UV. As reported in Figure 5, the SHR was dose-dependently reduced by bestatin.

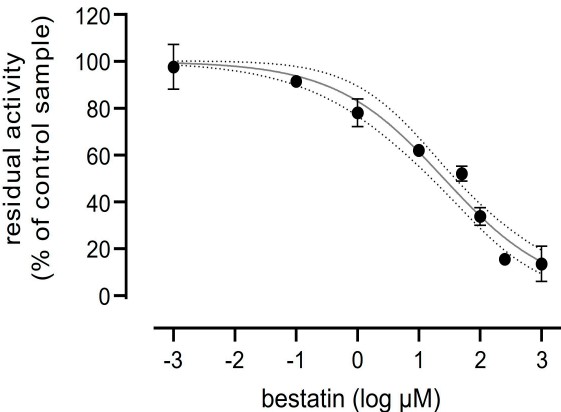

**Figure 5.** The effect of bestatin on serum hydrolytic activity as measured by IC-UV (average ± standard deviation of four independent measures). Fitted curve (solid line) with 95% confidence bands (dashed lines).

The simplest model describing the relationship between the concentration of bestatin in serum (X) and the percent of residual hydrolytic activity (SHR%) was a dose–response inhibition curve with a variable slope (S) described by the following equation:

$$SHR\% = 100/(1 + (IC50/X)\hat{\ }S) \tag{4}$$

The curve fit data with an adjusted R-squared value of 0.96. The concentration of bestatin that decreased the SHR to 50% (i.e., IC50) was 25 µM (95% confidence interval 16–39 µM). The extra-sum-of-squares F test rejected the null hypothesis that a simplest model without variable slope (i.e., S = 1) equally fit the experimental data.

The slope (S) of the best fitting curve was −0.5, which is the expected slope for enzyme kinetics where some catalytic activity is also retained by the complex between the enzyme (E) and the inhibitor (I) [30]. This is in line with CN1's structure, being a homo-dimeric enzyme with two catalytic pockets working independently and alternatingly [31]. This implies that the occupation of one pocket produces an enzyme–inhibitor complex (EI) that is still able to bind the substrate (S) and convert it into the product (P):

$$E + I \rightarrow EI + S \rightarrow EIS \rightarrow EI + P \tag{5}$$

To completely inhibit CN1, it is therefore necessary to occupy both binding pockets. This requires the binding of two molecules of the inhibitor:

$$EI + I \rightarrow E(I)_2 + S \rightarrow no\ binding \tag{6}$$

## 4. Conclusions

Herein, a chromatographic method is reported for the measurement of serum hydrolytic activity operated by the enzyme CN1 on different substrates (i.e., carnosine and homocarnosine). The method was based on the separation and quantification of histidine using ion chromatography with ultraviolet spectrophotometric detection (IC-UV). The results were consistent with the literature, especially when compared to an alternative chromatographic method based on hydrophilic interaction liquid chromatography with mass spectrometric detection (HILIC-MS). HILIC-MS surely performs better than IC-UV in terms of sensitivity (e.g., LLOQ and LOD) and selectivity (thanks to MS selective detection). However, the scope of method validation is to make sure that it is suitable for the intended purpose. In the specific case herein reported, we wanted to apply the method for the measurement of the hydrolysis of carnosine (or homocarnosine) operated by serum CN1 enzyme. In this context, we demonstrated that IC-UV has sufficient sensitivity and selectivity to accomplish such a task. The IC-UV method also provided solid data on CN1

inhibition, as demonstrated by experiments performed using a known inhibitor of CN1 (i.e., bestatin). The only apparent limit of IC-UV is that when an inhibitor is tested the molecule must interfere with histidine separation and detection by UV, as in the case of bestatin. This is not a prerequisite of the HILIC-MS method since it can rely on the selectivity of the MS detector to isolate the signal of the target analyte from the noise, even in the case of poor chromatographic separation. Besides this potential limitation of use, the IC-UV method looks very convenient if compared with the HILIC-MS method. IC-UV analyses require cheaper instrumentation and can be accomplished without the use of organic solvents. Moreover, IC-UV analysis also saves time since it requires one injection per sample, while HILIC-MS requires two injections per sample, to determine the serum hydrolysis rate. We estimated that IC-UV is 20-fold cheaper than the HILIC-MS method (reagent cost per run). Considering that HILIC-MS requires two runs per sample and that the cost of disposing organic solvents was not included in the evaluation, the cost-effectiveness of the IC-UV method is further increased. Overall, IC-UV looks, therefore, as a cost-effective and green alternative to study CN1 activity and test molecules designed to inhibit such an enzyme.

**Author Contributions:** Conceptualization, L.R.; methodology, L.R.; validation, L.R.; formal analysis, L.R.; investigation, C.d.G.; writing—original draft preparation, L.R.; writing—review and editing, L.R.; supervision, L.R. All authors have read and agreed to the published version of the manuscript.

**Funding:** This research received no external funding.

**Institutional Review Board Statement:** Not applicable.

**Informed Consent Statement:** Not applicable.

**Data Availability Statement:** The data presented in this study are available on request from the corresponding author.

**Conflicts of Interest:** The authors declare no conflict of interest.

**Sample Availability:** Samples of the compounds are not available from the authors.

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
