# Peer review of "A Green and Cost-Effective Chromatographic Method for the Measurement of the Activity of Human Serum Carnosinase"

_separations, doi:10.3390/separations10080460_

Round 1

Reviewer 1 Report

The manuscript “A green and cost-effective chromatographic method for the measurement of the activity of human serum carnosinase” by Chiara de Giacomi and Luca Regazzoni reports on a targeted chromatographic method that allows for the detection of different substrates (i.e., carnosine and homocarnosine) of the enzyme CN1. A simple method using IC-MS with an isocratic solvent mixture and UV detection of the target compounds is compared to the more complex and more expensive HILIC-LCMS. The authors show sufficiently the effectiveness of the method, the manuscript is well written and the cost-effectiveness and avoidance of organic solvents deserves to be highlighted.

It would be interesting to know how large the analytic field is that works on carnosine and homocarnosine detection to understand a possible impact of the manuscript.

Figures 1, 3 and 4 would benefit from an assignment of the major peaks within the spectra.

Author Response

Response to reviewer's comments is included in the attached pdf file

Reviewer 2 Report

Reviewer report on manuscript Separations-2554756

The submitted experimental work presents data on the development and application of a green HPLC method for the determination of the activity of human serum carnosine.

In general terms the manuscript is well-written and figured. I agree with the authors’ claims about the novelty of their approach. However, the analytical figure of merits (LOD, LOQ, etc) of the proposed method and the previous HILIC-MS method of the authors Ref[4]) should be compared and discussed in the text. How the proposed IC-UV method could be compared with HILIC-MS in terms of LOD?

Based on the above statements, I recommend acceptance of the proposed method after revision.

Comments

1)     Line 66-67: correct to “o-phthalaldehyde”

2)     Line 114: correct as “10-fold”

3)     Line 155: what is the unit of centrifugation speed of 14000? It should be provided in the text.

4)     Line 121: correct to “5-fold”

5)     Line 136: correct to "flow rate”

6)     Line 151: revise as “t-test”

7)     Section 2.4 title: the information related to the data analysis should be moved to previous section (2.3) and the details about the method validation should be remained in section 2.4. ON this basis the title of section 2.4 should be revised as “Method validation”.

the English language is acceptable

Author Response

(The authors gave the same response as above.)

Reviewer 3 Report

The manuscript should be improved according to the following observations:

1. Chromatographic separation: An average value of 1.2 was obtained for resolution. According to Ph.Eur. (2.2.46. CHROMATOGRAPHIC SEPARATION TECHNIQUES) and also USP and JP (chapter undergone pharmacopoeial harmonization), the peak-to-valley ratio may be employed as a system suitability criterion when baseline separation between 2 peaks is not achieved.  Moreover, according to EDQM Technical guide for the elaboration of monographs, peak-to-valley ratio can be employed when complete separation between two adjacent peaks cannot be achieved (i.e. when the resolution is less than 1.5) and the minimum requirement for peak-to-valley ratio should not be less than 1.5. Therefore, please consider to evaluate the separation of histidine and carnosine by peak-to-valley ratio instead of resolution.

2. Line 157: The sentence must be revised for typos.

3. Lines 258, 262: Please confirm if reference to Figure 2B is correct. Reference to Figure 2A seems to be more plausible.

4. Lines 365-366: Please confirm if the sentence “The only apparent limit of IC-MS is that when an inhibitor is tested the molecule must interfere with histidine separation and detection by UV as in the case of bestatin.” is correct. Consider if „IC-UV” should be use instead of „IC-MS” and “…must not interfere...” instead of „...must interfere...”.

Author Response

(The authors gave the same response as above.)

Round 2

Reviewer 3 Report

The manuscript has been sufficiently improved.